# Youth-friendly health service in Ethiopia: Assessment of care friendliness and user's satisfaction

**Andamlak Gizaw Alamdo** [1,2]*, **Fanna Adugna Debelle**[1], **Peter Muriuki Gatheru** [2], **Adom Manu**[2], **Juliana Yartey Enos**[2,3], **Tewodros Getinet Yirtaw**[4]

**1** Department of Health Service Management, Health Promotion, Reproductive Health, and Nutrition, School of Public Health, Saint Paul's Hospital Millennium Medical College, Addis Ababa, Ethiopia, **2** Department of Population, Family and Reproductive Health, School of Public Health, College of Health Sciences, University of Ghana, Legon, Ghanaffi, **3** Department of Epidemiology, Noguchi Memorial Institute for Medical Research, College of Health Sciences, University of Ghana, Legon, Ghana, **4** Department of Epidemiology, School of Public Health, Saint Paul's Hospital Millennium Medical College, Addis Ababa, Ethiopia

* gizandal@gmail.com

## Abstract

### Background

Health facilities' level of readiness to provide adolescent and youth-friendly health services (AYFHS) is crucial for preventing and reducing teenage pregnancies and maternal mortality. This study examined friendliness and satisfaction with AYFHS among users.

### Methods

A cross-sectional study was conducted among 419 adolescents and youths recruited from six health institutions in the East Shewa Zone, Oromia region, and Addis Ababa, Ethiopia from August 1 to October 28, 2022. Based on the WHO Quality Assessment Guidebook, the friendliness of AYFHS was evaluated as a proxy for the quality of care. Descriptive measures were computed to summarize the participants' socio-demographic characteristics. Bivariate and multivariable logistic regression analysis was used to evaluate the potential association between each independent variable and client satisfaction. The type, magnitude, and precision of association were presented using an adjusted odds ratio (AOR) with 95% Confidence Intervals (CI) and a p-value of < 0.05 ascertained statistical significance.

### Results

The overall friendliness was 69%: Specifically, 82% of the participants stated the service was accessible, 72% said it was acceptable, and 90% thought it was effective. However, only 33% and 34% of respondents reported the services were equitable and appropriate respectively. The overall satisfaction with AYFHS was 25.3%. Increased age (AOR = 0.70, 95% CI: 0.57–0.86), being female (AOR = 0.11, 95%CI: 0.04–0.33), no prior information about AYFHS (AOR = 0.20, 95%CI: 0.10–0.44), and higher number of sexual partners (AOR = 0.3, 95%CI: 0.21–0.65) were factors statistically associated with client satisfaction with AYFHS.

**Data Availability Statement:** All relevant data are within the manuscript and its Supporting Information files.

**Funding:** The study was sponsored by St. Paul's Hospital Millennium Medical College Research and Resources Mobilization Directorate. The funders had no role in study design, data collection and analysis, decision to publish, or preparation of the manuscript.

**Competing interests:** The authors have declared that no competing interests exist.

## Conclusions

The overall quality of AYFHS was sub-optimal and did not meet the four components of WHO's good quality standards; equity, appropriateness, acceptability, and accessibility must be improved. Female and older adolescents and youths and those with a higher number of sexual partners should be targeted for intervention.

## Introduction

Adolescents (10–19 years) and youth (15–24 years) face particular challenges in acquiring services related to sexual and reproductive health [1]. The lack of access to health information and services has resulted in risky sexual behaviours that have increased the prevalence of STIs and HIV and contributed to early pregnancy, susceptible to delivery complications, and high rates of death and disability. In Ethiopia, approximately 45% of the population is under 30 years of age. The greatest number of Ethiopians to reach adulthood in history are young people between the ages of 10 and 24 [2, 3].

In Ethiopia, 14% of females are married before age 15, and 40% get married before turning 18 [4]. Nationally, 13% of females aged 15–19 have already given birth, and 2% are pregnant with their first child [5]. Moreover, only a quarter 23.8% of married females aged 15–19 use any method of contraception. Of young people aged 15–24, 75% do not have comprehensive knowledge about HIV [6].

Adolescent-friendly health services are designed to meet the unique needs of teenagers, providing them with confidential, non-judgmental, and comprehensive care [7]. These services recognize adolescents' developmental, social, and emotional challenges and aim to create a supportive environment where they feel comfortable seeking help and information [7, 8]. Adolescent and youth-friendly health services (AYFHS) can improve access to health services and improve health outcomes for young people [9]. According to WHO, AYFHS are accessible, acceptable, and appropriate in terms of the right place, time, and affordability [9]. Overall quality of Adolescent and youth-friendly health services can influence the satisfaction levels of young people. Young people's satisfaction with youth-friendly healthcare is a measure of the extent to which they feel that their healthcare needs are being served in a professional, effective, and useful way. It includes a range of aspects related to their experience, including as accessibility, level of care, privacy, communication with healthcare providers, and the overall environment of the healthcare facility [8, 10]. Low satisfaction may lead to a decrease in the utilization of health services by adolescents and young people. Additionally, young people may avoid seeking care altogether or delay seeking care until their health condition worsens, leading to poorer health outcomes [8]. Moreover, low satisfaction can result in missed opportunities to address preventive healthcare needs and may result in increased risk factors for various health conditions [10, 11].

In the past two decades, the government of Ethiopia has developed several policies and regulations to help implement AYFHS. Ethiopia's first Adolescent and Youth Reproductive Health Strategy was developed in 2006 and put into practice from 2007 to 2015 [12]. A National Adolescent and Youth Health strategy was developed in 2016 in response to the ongoing changes in the epidemiological profile of adolescents [5]. It is believed that the strategy will enhance and sustain the mainstreaming of adolescents and youth health and rights issues into the country's growth and transformation agenda.

Although Ethiopia has a national strategy for the reproductive health of adolescents and young people and a cadre of health extension workers (HEWs) who provide primary preventive healthcare at the community level, access to and experience with these health services is limited [13]. Despite the overwhelming load of SRH problems, the adolescent SRH services are still not adequately staffed with qualified professionals, allocated space or time. With little emphasis on services for adolescents and youths, most healthcare facilities provide SRH services as part of standard medical care [13].

To reduce and prevent unintended teenage pregnancies and maternal mortality, which are unacceptably high in Ethiopia, health facilities' level of preparedness to provide ASRH services is crucial to the extent to which young people use these services [14]. Even though AYFHS has been providing care in Ethiopia for nearly 20 years, there is limited evidence on the level of care that meets international standards. Additionally, the existing literature focuses mainly on examining factors that affect AYFH services utilization and examining quality of care concentrating only on a particular aspect. Therefore, this study provides local, context-specific evidence about the degree of friendliness provided by adolescent and youth health services in Ethiopia by using the WHO Quality Assessment Guidebook [15]. In addition, the study presented the SRH exposure of adolescents and their satisfaction with the services provided in Addis Ababa city administration and the East Shewa zone in the Oromia region, Ethiopia.

## Methods

### Study design, settings and period

A cross-sectional study was conducted to examine the AYFHS provided by Addis Ababa city administration and the East Shewa zone in the Oromia region, Ethiopia. Addis Ababa is the capital and largest city of Ethiopia. The city is divided into three administrative tiers: one government, 10 sub-cities, and 116 woreda administrations. According to the World Population Review, the population of Addis Ababa, Ethiopia, is estimated to be 5.7 million people by 2024. The gender distribution of the population of Addis Ababa shows that women account for 52.4% and males make up around 47.6%. Data indicate that about 2.28 million residents are likely to be under 15 years old [16]. Moreover, according to recent estimates, the East Shewa Zone of Oromia, Ethiopia, is home to about 1.7 million people [17]. During the selection of the study areas, the presence of health facilities rendering AYFHS and accessibility of the health facilities were considered as criteria. Addis Ababa is the capital city of Ethiopia; AYFHS is relatively more accessible to the young population. In addition, Addis Ababa city has functioning youth-friendly health service facilities that provide comprehensive sexual and reproductive health services, including counselling, HIV testing, contraceptive services, STI diagnosis, and management. East Shewa zone is one of the administrative zones in the Oromia Regional state, for this study, it represents areas out of the capital where AYFHS is assumed less accessible. In selecting the East Shewa zone, health authorities' input, historical performances of local health facilities in the provision of health services, geographical variety to capture variation in the quality of health services, and feasibility and resources allocated for the study were taken into account. Addis Ababa has eleven administrative subdivisions, called sub-cities. Each sub-city has at least one health facility providing AYFH services. From Addis Ababa city administration, Yeka Health Center, Yekatit Hospital, Gulelle Health Center, Nefas Silk Lafto wereda 2 Health Center, and Nefas Silk Lafto wereda 6 Health Center were selected for the study. Furthermore, Bosset Health Center from the East Shewa Zone of the Oromia region, Ethiopia. The study was conducted from August 1 to October 28, 2022.

## Study participants and selection procedure

All adolescents (10–19 years) and youths (15–24 years) in the Addis Ababa city administration and East Shewa Zone of Oromia region, Ethiopia were the target population. Bosset health center from the East Shewa zone was selected by considering the presence of a youth-friendly corner within the facility, safety, and security to travel the facility. All adolescents (10–19 years) and youths (15–24 years) who visited AYFH services during the study period, in selected health facilities in the study areas, were the study population. Adolescents and youths who utilized AYFH services and lived in the study area for at least six months were eligible to participate in the study. Adolescents and youths with emergency medical issues were excluded from the study.

## Sample size and sampling technique

The required sample size for the quantitative study was determined using the single population proportion formula. Taking 50% of adolescents and youths satisfied with AYFH services provided (as there was no published study), $Z\alpha/2 = 1.96$ for a 95% level of confidence, and d = 5% required level of precision. Therefore, the sample size (n) = 424 was considered after adding 10% for a possible nonresponse rate. There are a total of 86 public health facilities in Addis Ababa of which 11 health facilities are providing AYFHS in Addis Ababa City Administration. From the existing 11 health facilities that deliver AYFHs, five were selected using the lottery method of simple random sampling technique. The health facilities selected in Addis Ababa were Yekatit 12 Hospital, Selam HC, Nefas Silk Lafto Wereda 2 HC, and Nefas Silk Lafto wereda 6 HC. Bosset HC was selected from the East Shewa zone, Oromia region. The adolescents and youths were selected with probability proportional to the number of adolescent and youth clients that each healthcare facility served in the previous year to the start of the study. The study participants were then selected in every other client exit by using systematic random sampling techniques.

## Data collection tools and procedures

Data were collected using computer-assisted face-to-face interviews using the Open Data Kit (ODK). The tool was initially developed in English and translated into the local languages (Amharic and Afaan Oromo), and then back-translated to the English language by the language expert, to make sure of its consistency and accuracy. The tool was developed, based on the review of relevant literature [3, 15, 18, 19], to gather information from AYFHS clients on their sociodemographic profiles (age, sex, marital status, educational status, etc.), SRH history (sexuality, pregnancy, contraception use, etc.), AYFH program/service exposure, perception of care friendliness, and satisfaction (S1 File). The WHO Quality Assessment Guidebook: a guide to evaluate health services for adolescent clients was followed as a guideline [15]. Six data collectors with a BSc degree in public health and Midwifery and who have experience working with the young population interviewed the study participants. Moreover, two supervisors with master's degrees in public health (MPH) closely supervised the data collectors in the field. All the data collectors and supervisors were trained on the objectives of the study, ethical issues, the procedures of data collection, and data collection tools.

## Study variables

Existing patterns of adolescent youth-friendly health services (AYFHS) were evaluated based on clients' exit interviews in selected health facilities. Care friendliness was measured using 19 questions (characteristics) which were grouped into five domains: equity (one characteristic),

accessibility (seven characteristics), acceptability (five characteristics), appropriateness (four characteristics), and effectiveness (three characteristics) (S1 Table).

- Utilization of reproductive health services: refers to the use of reproductive health services including medical check-ups, consultations, Family Planning, health education on HIV/AIDS, and STI treatment services provided in health centers.

- Hours for young people: refers to the AYFRH unit was opened at times that are convenient for young people to attend. Such times include late afternoons (after school or work), evenings, and weekends.

- Convenient location: refers to AYFRH unit was easily accessible by foot or public transportation

- Satisfied: the respondent scored $\geq 6$ from 11 points on client satisfaction tools and was satisfied with three items waiting time to get the service, privacy, and feeling comfortable with the services provided [20].

- To be considered adolescent-friendly, services should have the following characteristics [15].

  **Equitable.** All adolescents, not just certain groups, can obtain the health services they need.
  **Accessible.** Adolescents can obtain the services that are provided.
  **Acceptable.** Health services are provided in ways that meet the expectations of adolescent clients.
  **Appropriate.** The health services that adolescents need are provided.
  **Effective.** The right health services are provided in the right way and make a positive contribution to the health of adolescents.

- Care friendliness: we categorized the index of expected friendliness in AYFH services for adolescents into three categories: low ($<0.80$), moderate ($0.80–0.89$), and high ($\geq 0.90$) [19].

## Data quality control

Qualified personnel who have experience in quantitative data collection and interview techniques were deployed to collect the data. Data collectors and supervisors were trained to make each person thoroughly familiar with data collection instruments & procedures. Before the actual data collection pre-test was conducted, in health facilities that were not part of the study area, to check for language clarity and response appropriateness. During data collection in the field, data were checked for completeness and accuracy before leaving the data collection site. A standardized WHO quality assessment guide was used to examine care friendliness [15]. Using Cronbach's alpha reliability test, the overall internal reliability of the survey instrument for assessing client satisfaction was examined, and the scores for the items of each component were more than 0.71.

## Data processing and analysis

Before data analysis, data cleaning was carried out through a series of cross-tabulations, frequency tables, and raw data check-ups. Stata16 (Stata Corp, College Station, Texas, USA), was used for all data analysis. Descriptive measures were computed to summarize the participants' socio-demographic characteristics, and the friendliness of AYFHS, a proxy for quality of care was assessed based on the WHO Quality Assessment Guidebook [15]. To calculate the relative score by characteristic, we first determined the percentage of positive answers for each question on the friendliness of care. We then combined the percentages of each question by domain and divided this by the total number of questions to derive a total for each domain within the same

range: from zero to one. Lastly, we computed the global score by dividing the total number of questions by the sum of all positive responses for friendliness, following the methodology for each domain that was recommended by the World Health Organisation [15]. The study used both bivariate and multivariate logistic regression analysis to assess the potential association between each independent variable and client satisfaction. Multiple variable logistic regression analysis was carried out using independent variables that showed a p-value of less than 0.2 in the bi-variable logistic regression. At last, the magnitude and precision of association were presented using an AOR with 95% CI, and a p-value of $< 0.05$ ascertained statistical significance.

### Ethical consideration

The proposal was approved by the Institutional Review Board (IRB) of St. Paul's Hospital Millennium Medical College (SPHMMC) (Ref. No: P.M23/502/31/03/2022). In addition, before contacting participants, permission was obtained from the Addis Ababa and Oromia regional state health bureaus, and the respective district health offices and health facilities. Verbal informed consent was sought before any data were collected. Asking for written informed consent could cause discomfort for the study participants, who visited the health facilities with caution to avoid getting caught by someone they knew. Moreover, the SPHMMC IRB typically requires verbal informed consent for research that is not clinical trials involving invasive procedures. Hence, verbal informed consent was requested, and moving on to the next set of questions assumes that the adolescents and youths have given their consent to participate. Many adolescents and youths aged less than 18 years utilize AYFH services without consulting their parents/guardians and asking for parental consent might cause emotional harm to adolescents and youths. Due to this, parents' consent was not sought, and the IRB of SPHMMC waived it. Participants in the study were given enough information regarding the study's purpose, objective, benefits, and harm. Participants were also informed that their participation is fully voluntary and they have the right to withdraw and jump any question. Privacy and confidentiality were maintained, respectively, by not exposing individual identifiers, not transferring the data to a third party, and using it for research purposes only.

## Results

### Socio-demographic characteristics

A total of 419 adolescents and youths, 319 (76.1%) from Addis Ababa and 100 (23.9%) from the East Shewa zone of the Oromia region took part in the study, yielding a 98.8% response rate. For the study, six health facilities that provide services related to adolescent and youth reproductive health were identified. Of these, 99 (23.6%) were from Yekatit 12 Hospital and nearly a quarter of 100 (23.9%) were from Bosset Health Center. Participants' median age was 21 (IQR 5), and 340 (81.2%) of them were female, a fairly large proportion. Moreover, the minimum age at first marriage was 15 while the maximum was 24 years and the mean age at first marriage was 19.5 (SD ± 2.0). A large majority 297 (70.9%) of participants were single. Regarding religious affiliation, three-fourths (74.9%) of the study participants were Orthodox Christian followed by Muslim 57 (13.6%). Less than half 170 (40.6%) of the participants reported that they were in school and over a third 147 (35.1%) attended at least primary school (grade 1–8) (Table 1).

### Sexuality, pregnancy, and contraception history of study participants

In this study, eighty percent 336 (80.2%) of the study participants ever had sexual intercourse. Of these, 289 (86%) had sexual intercourse in the last six months before the survey period. For those who had started sexual intercourse, the minimum age at first sex was 11 years while the

**Table 1. Demographic and socio-economic background of study participants in East Shewa Zone, Oromia region, and Addis Ababa Ethiopia, October 2022 (n = 419).**

| Variable | Category | Number | Percent |
|---|---|---|---|
| Region | Addis Ababa | 319 | 76.1 |
| | East Shewa Zone, Oromia | 100 | 23.9 |
| Health facility | Addisu Gebeya HC | 62 | 14.8 |
| | Nifas Silik Lafto wereda 2 HC | 73 | 17.4 |
| | Nifas Silik Lafto wereda 6 HC | 41 | 9.8 |
| | Bosset HC | 100 | 23.9 |
| | Selam HC | 44 | 10.5 |
| | Yekatit 12 Hospital | 99 | 23.6 |
| Sex | Female | 340 | 81.2 |
| | Male | 79 | 18.9 |
| Age | 19 years or less | 130 | 31.0 |
| | 20–24 years | 289 | 68.9 |
| Religion | Orthodox | 314 | 74.9 |
| | Muslim | 57 | 13.6 |
| | Protestant | 42 | 10.0 |
| | Catholic | 4 | 1.0 |
| | Other* | 2 | 0.5 |
| Marital status | Single | 297 | 70.9 |
| | Married | 114 | 27.2 |
| | Divorced/Separated | 8 | 1.9 |
| Educational level | Primary Level (1–8) | 147 | 35.1 |
| | Secondary Level (9–12) | 142 | 33.9 |
| | Over Secondary Level | 122 | 29.1 |
| | Other** | 8 | 1.9 |
| Current Educational Status | In-School | 170 | 40.6 |
| | Out-of-School | 249 | 59.4 |

*No religion, Waqefeta

**No education

HC, health center

mean age at first sex was 18 (SD ± 2.0). In addition, the minimum number of sexual partners reported was 1 while the maximum was 18 and the median number of sexual partners were 2 with IQR (1 3). A total of 204 (70.6%) adolescents and youths utilized a method to prevent pregnancy or disease. Among these, nearly a third of the participants used pills 69 (33.8%) and injectables 62 (30.4%). When asked about the contraceptive method to be taken within 72 hrs. to prevent unwanted pregnancy, a considerable number 272 (64.9%) reported emergency contraceptives while nearly one-third 134 (31.9%) of them didn't respond/couldn't mention one.

The study also revealed the minimum age at first pregnancy to be 14 years and the median age at first pregnancy was 20 (IQR 2). Just over forty percent 112 (41%) of female participants had ever been pregnant. Of these, more than half 63 (56.3%) gave birth while nearly a quarter 27 (24.1%) experienced induced abortion. Twenty-one (77.8%) participants who faced induced abortion reported private clinic 21 (77.8%) as the place of induced abortion. Regarding the history of contraception use, 191 (70%) female participants reported the use of any contraception method. Among these, 93 (48.7%) got contraceptive methods from public health facilities and 61 (31.9%) from pharmacies (Table 2).

**Table 2. Study participants' history of sexuality, pregnancy, and contraception in East Shewa Zone, Oromia region, and Addis Ababa Ethiopia, October 2022 (n = 419).**

| Variables | Category | Number | Percent |
|---|---|---|---|
| Have you ever had sexual intercourse? (n = 419) | Yes | 336 | 80.2 |
| | No | 83 | 19.8 |
| Had sexual intercourse in the last six month? (n = 336) | Yes | 289 | 86.0 |
| | No | 47 | 14.0 |
| Did you use a method to prevent pregnancy or disease? (n = 289) | Yes | 204 | 70.6 |
| | No | 40 | 13.8 |
| | Don't remember/No response | 45 | 15.6 |
| Which method(s) did you use? (n = 204) | Condom | 28 | 13.7 |
| | Pills | 69 | 33.8 |
| | Injectables | 62 | 30.4 |
| | IUD/Loop | 23 | 11.3 |
| | Norplant | 48 | 23.5 |
| | Safe days/Rhythm | 3 | 1.5 |
| | Withdrawal | 1 | 0.5 |
| | Other* | 3 | 1.5 |
| Contraceptive method to be taken within 72 hrs to prevent unwanted pregnancy? (n = 419) | Emergency Contraceptive | 272 | 64.9 |
| | Condom | 4 | 1.0 |
| | Injectables | 9 | 2.2 |
| | Don't know/No response | 134 | 31.9 |
| Ever been pregnant? (n = 273) | Yes | 112 | 41.0 |
| | No | 161 | 59.0 |
| How did the pregnancy end? (n = 112) | Gave birth | 63 | 56.3 |
| | Still birth | 16 | 14.3 |
| | Spontaneous abortion | 6 | 5.4 |
| | Induced abortion | 27 | 24.1 |
| Place of induced abortion (n = 27) | Project HF | 6 | 22.2 |
| | Private clinic | 21 | 77.8 |
| Do you use contraceptive? (n = 273) | Yes | 191 | 70.0 |
| | No | 82 | 30.0 |
| Where did you collect the contraceptive? (n = 191) | Project HF | 93 | 48.7 |
| | Private clinic | 35 | 18.3 |
| | Pharmacy | 61 | 31.9 |
| | Other** | 2 | 1.1 |

*Emergency contraceptive pills

** Hospitals and health center outside their catchment

HF, health facility

IUD, Intrauterine device

## AYFH program/service exposure of study participants

Only about half 219 (52.3%) of adolescents and youths in this study had prior information about the presence of AYFHS in the health facilities. Of these, a considerable number 137 (62.6%) got the information from their friends followed by healthcare personnel 49 (22.4%). When asked what services were provided in the HF, a very large majority 158 (72.2%) responded comprehensive contraception followed by STIs diagnosis and treatment 115 (52.2%). Among the study participants who had information about AYFHS, almost all

participants 198 (90.4%) ever visited AYFHS service in the HF and the main reason for their visit was comprehensive contraception 82 (41.4%) and pregnancy test 42 (21.2%). When asked about any obstacles to getting AYFH services, 61 (16%) participants reported there were challenges to getting the services. Of these, an overwhelming number 53 (86.9%) mentioned lack of privacy and confidentiality, fear of being seen by family/neighbours 26 (42.6%), and working hours are not convenient 16 (26.2%). Concerning the recommended arrangements for the AYFH services, 344 (82.7%) reported special rooms, recreational sites 218 (52.4%), and special times 124 (29.8%) (Table 3).

## Level of friendliness with AYFHS from the users' perspective

As shown in Table 4 and Fig 1 below, 82% of the participants stated the service was accessible, 72% reported it was acceptable, and 90% thought the service was effective. However, only 33% and 34% of respondents reported the services were equitable and appropriate respectively. Specifically, about a third of adolescents and youths (66%) believed that there were no policies and procedures in place that ensure equitable health care. The lowest reported appropriateness characteristics were the availability of peer support/mentorship (24%) and youth-friendly educational activities that address topics based on adolescent's interests (24%). Additionally, only 31% of adolescents and youths stated the services were advertised in places where they congregate.

In comparison, an overwhelming proportion of (82%) adolescents and youths believed that service was accessible in terms of service affordability/ services free of cost (100%) and the point of health service delivery has convenient hours of operation (99%). Despite this, only 27% of the participants were well-informed about the range of available health services and referral linkage with other youth-friendly or social services in the community (Table 4).

## Client satisfaction with adolescent and youth friendly health services

In this study, the level of client satisfaction with the service provided at adolescent and youth units was assessed using eleven characteristics of care. Accordingly, the overall satisfaction with AYFH service in the study area was 68.9%. Besides this, the overall client satisfaction was also evaluated using three mandatory criteria: waiting time to receive the service, privacy, and feeling comfortable with the services offered, and satisfaction with at least six items out of eleven characteristics of care to categorize adolescents and youths as satisfied with AYFHS or not. This led to the documentation of a low overall client satisfaction rate of 25.3%.

Concerning the characteristics of care, nearly all adolescents and youths were satisfied with service accessibility (96.9%), service affordability (99.8%), and the welcoming physical facility (99.8%). Similarly, almost all (98.9%) would recommend the health site to a friend who needed similar help. While 92.4% felt comfortable talking to all of the people working at the health facility, and 94.8% believed that everything they discussed with healthcare providers would remain confidential. A considerable number of study participants (63.5%) were satisfied with the privacy in the treatment/consultation room. However, only a third (35.1%) of adolescents and youths were satisfied with the acceptability of the waiting time and just over thirty percent (30.1%) of and less than thirty percent (27.9%) were satisfied with referral linkage and provider-client interaction respectively. Furthermore, only 19.8% of adolescents and youths felt well with the healthcare facility's service providers (Fig 2).

## Factors associated with adolescent and youth client satisfaction

In Bivariate analysis, AYFHS satisfaction was significantly associated with age, educational status, current educational status, marital status, number of sexual partners, and AYFH service

**Table 3. Study participants' exposure to AYFH program or services in East Shewa Zone, Oromia region, and Addis Ababa Ethiopia, October 2022 (n = 419).**

| Variables | Category | Number | Percent |
|---|---|---|---|
| Have information about AYFHS service in the facility? (n = 419) | Yes | 219 | 52.3 |
| | No | 200 | 47.7 |
| Source of information? (n = 219) | Friends | 137 | 62.6 |
| | Health personnel | 49 | 22.4 |
| | Teachers | 1 | 0.5 |
| | Parents | 24 | 10.9 |
| | Public announcement | 8 | 3.7 |
| What services are provided in the HF*? (n = 219) | Comprehensive abortion care | 81 | 36.9 |
| | Comprehensive contraceptive | 158 | 72.2 |
| | Pregnancy test | 112 | 51.1 |
| | STIs diagnosis and treatment | 115 | 52.5 |
| | Other* | 7 | 3.2 |
| Ever visited the AYFH service in the HF? (n = 219) | Yes | 198 | 90.4 |
| | No | 21 | 9.6 |
| Reason for your visit? (n = 198) | Comprehensive abortion care | 25 | 12.6 |
| | Comprehensive contraceptive | 82 | 41.4 |
| | Pregnancy test | 42 | 21.2 |
| | STIs diagnosis and treatment | 33 | 16.7 |
| | Other** | 83 | 41.9 |
| Any obstacles to get AYFH services? (n = 419) | Yes | 61 | 14.6 |
| | No | 358 | 85.4 |
| What are the obstacles? (n = 61) | HF too far | 10 | 16.4 |
| | Fear of seen by family/neighbours | 26 | 42.6 |
| | Lack of privacy and confidentiality | 53 | 86.9 |
| | Providers' judgmental attitude | 4 | 6.6 |
| | Waiting time is too long | 1 | 1.6 |
| | Working hours are not convenient | 16 | 26.2 |
| | Other*** | 1 | 1.6 |
| | Don't know/No response | 1 | 1.6 |
| Any recommended arrangement for AYFH services? (n = 419) | Yes | 416 | 99.3 |
| | No | 3 | 0.7 |
| What are the recommended arrangements for AYFH services? (n = 416) | Special room | 344 | 82.7 |
| | Special time | 124 | 29.8 |
| | Weekend service | 119 | 28.6 |
| | Reading corner | 112 | 26.9 |
| | Recreational sites | 218 | 52.4 |
| Preference of AYFH service provider? (n = 419) | Yes | 418 | 99.8 |
| | No | 1 | 0.2 |

*Urine test, Advice, I don't know, Condom

**other medical services

*** lack of information

AYFHS, adolescent and youth-friendly health services; HF, health facility

information. After adjustment for confounding factors, this study identified four variables that had statistically significant associations with adolescent and youth-friendly health services satisfaction. Accordingly, female adolescents and youths were 89% less likely to be satisfied with AYFH services compared to males (AOR: 0.11, 95% CI: 0.04–0.33). Additionally, for a unit

**Table 4. Friendliness of AYFH services according to characteristics and domains in East Shewa Zone, Oromia region, and Addis Ababa Ethiopia, October 2022.**

| Domain | Characteristics | Score |
|---|---|---|
| Equitable | Are procedures in place to ensure that no young people are excluded from services? | 0.33 |
| Relative score | | 0.33 |
| Accessible | Is information and referrals provided about where young people can access other youth-friendly health or social services in the community? | 0.27 |
| | Have you found the waiting times too long before seeing the health-care providers? | 0.60 |
| | Are the working days and working hours of the health facility convenient for you? | 0.99 |
| | Are services located in an area that is accessible to youth and safe for them to travel to? | 0.99 |
| | Are services free of cost or affordable for young people? | 1 |
| | Are you able to access all of their health services in one visit? | 0.97 |
| | Are there separate clinic hours or waiting areas just for young people? | 0.9 |
| Relative score | | 0.82 |
| Appropriate | Does the site have posters, brochures and other IEC materials that target young people, including information about their rights? | 0.57 |
| | Are the services advertised to young people in places where they congregate (e.g., schools, youth clubs, recreation centers, etc.)? | 0.3 |
| | Is peer support or mentoring available? | 0.23 |
| | Are educational activities youth-friendly and address topics of interest to youth? (e.g., role plays, theatre, games, etc.) | 0.24 |
| Relative score | | 0.34 |
| Acceptable | Are youth involved in program design, delivery, and evaluation? | 0.27 |
| | Are young people greeted warmly upon entering? | 0.97 |
| | Are sessions conducted in an area that provides privacy so that nobody can see or hear the conversations taking place? | 0.64 |
| | Do you think that your parents/guardians would be supportive of you coming to this health facility for reproductive health services? | 0.99 |
| | Do you believe that the information you shared with the health-care provider will be kept confidential? | 0.73 |
| Relative score | | 0.72 |
| Effective | Are condoms/other methods available to young people on-site? | 0.72 |
| | Competent health-care providers | 0.98 |
| | Does the site have a youth-friendly strategy or action plan in place? | 0.99 |
| Relative score | | 0.90 |
| Friendliness | Global score | 0.69 |

increase in the age of adolescents and youth, the odds of satisfaction with AYFH services would decrease by 30% (AOR: 0.70, 95% CI: 0.57–0.86). Moreover, the study indicated that a higher number of sexual partners was associated with decreased odds of satisfaction with AYFH services (AOR: 0.37, 95% CI: 0.21–0.65). Furthermore, the odds of adolescents and youth who had no prior information about AYFH services had 80% lower odds of satisfaction with AYFH services (AOR: 0.20, 95% CI: 0.10–0.44) (Table 5).

## Discussion

This study presented the adolescents' and youth health service exposure of adolescents, care friendliness, and their satisfaction with the services provided in Addis Ababa city administration and East Shewa zone in the Oromia region, Ethiopia. According to the findings, the perceived quality of AYFHS from Adolescents' and youth's perspectives is sub-optimal (overall friendliness was 69%). Only one-third of adolescents and youths reported that the services

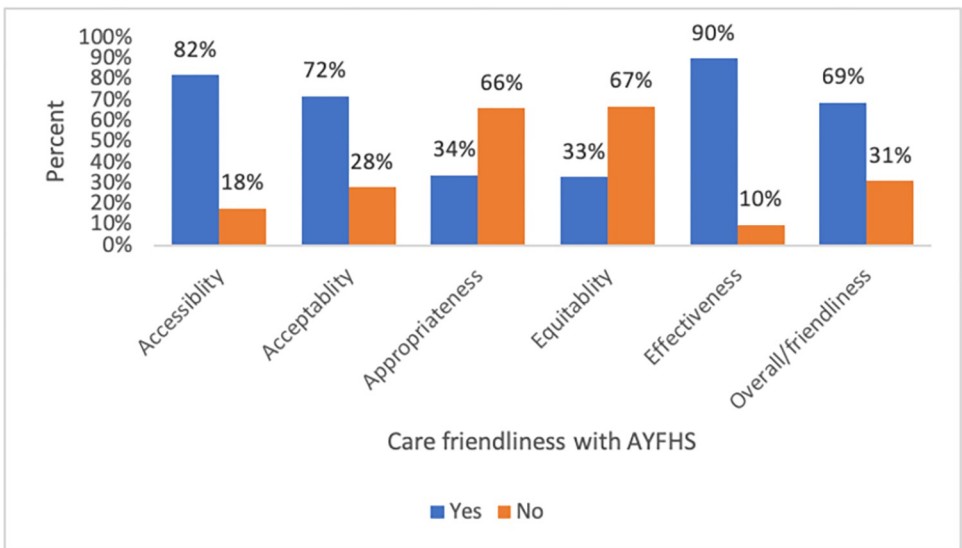

**Fig 1. Components of AYFHS from the client's perspective in East Shewa Zone, Oromia region, and Addis Ababa Ethiopia, October 2022.**

were appropriate (34%) and equitable (33%). Low appropriateness of service (34%) resulted from the scarcity of peer support groups in the health facilities, lack of youth-friendly educational activities that address topics based on adolescents' interests, and services not well advertised at young people gatherings such as schools, and youth clubs. These findings are similar to a study from Plateau State Nigeria where the overall SRH services in the state were not appropriate for adolescents [21]. This finding is also consistent with an assessment conducted using a similar methodology on youth-friendly health services in two South African provinces [22].

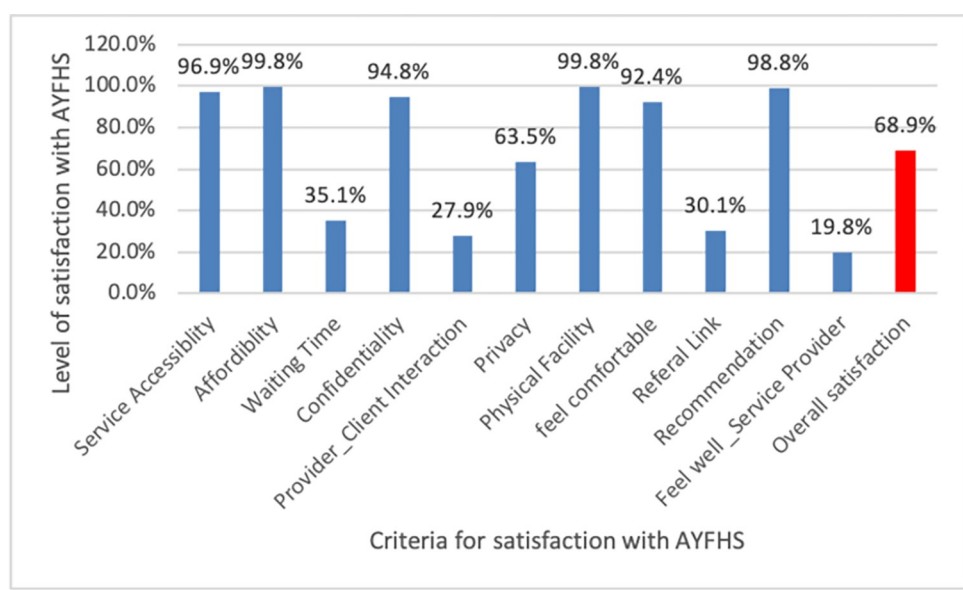

**Fig 2. Overall AYFH service satisfaction among adolescents and youths in East Shewa Zone, Oromia region, and Addis Ababa Ethiopia, October 2022.**

**Table 5. Factors associated with AYFH service satisfaction among adolescents and youths in East Shewa Zone, Oromia region, and Addis Ababa, Ethiopia, October 2022.**

| Variables | | AYFRH service satisfaction | | | COR (95% CI) | AOR (95% CI) |
|---|---|---|---|---|---|---|
| | | Yes (%) | No (%) | | | |
| Age | Mean age | 18.68 | 21.68 | | 0.64 (0.58–0.71)* | 0.70 (0.57–0.86)* |
| Sex | Male | 23 (29.11) | 56 (70.89) | | 1 | 1 |
| | Female | 83 (24.41) | 257 (75.59) | | 0.79 (0.46–1.36) | 0.11 (0.04–0.33)* |
| Educational status | Primary | 56 (38.10) | 91 (61.90) | | 1 | 1 |
| | Secondary | 38 (26.76) | 104 (73.24) | | 0.60 (0.36–0.98)* | 1.84 (0.75–4.54) |
| | Over secondary | 12 (9.84) | 110 (90.16) | | 0.18 (0.09–0.35)* | 1.33 (0.40–4.47) |
| | Other | 0 | 8 (100) | | 0 | 0 |
| Current Educational status | Currently in School | 61 (35.88) | 109 (64.12) | | 1 | 1 |
| | Out of school | 45 (18.07) | 204 (81.93) | | 0.39 (0.25–0.62)* | 2.19 (0.89–5.42) |
| Marital status | Single | 90 (30.30) | 207 (69.70) | | 1 | 1 |
| | Ever married | 16 (13.1) | 106 (86.9) | | 0.35 (0.19–0.62)* | 1.40 (0.58–3.40) |
| Number of sexual Partner | Mean | 1.35 | 2.67 | | 0.46 (0.28–0.70)* | 0.37 (0.21–0.65)* |
| AYFH service information | Yes | 83 (37.90) | 136 (62.10) | | 1 | 1 |
| | No | 23 (11.50) | 177 (88.50) | | 0.21 (0.13–0.36)* | 0.20 (0.10–0.44)* |

"1' indicates the reference/baseline category for binary and multivariable logistic regression analysis

AYFHS, adolescent and youth-friendly health services; COR: crude odds ratio; AOR: adjusted odds ratio; CI: confidence interval

*Represents statistically significant association

The study revealed that all primary care health facilities failed to meet the recommended standards for providing adolescent and youth-friendly health services.

One of the criteria for youth-friendly health services is all adolescents should be able to obtain the health services they need. Moreover, the existence of policies and procedures to ensure equitable health services is crucial [15, 19]. Despite this, the current study demonstrated inequitable health services (67%) due to a lack of procedures to ensure no young people are excluded from services. This calls for the implementation of process guidelines for comprehensive adolescent health as well as the updating of such policies and procedures [23].

Even though nearly three-fourths (72%) of the study participants reported services were acceptable, the active participation of adolescents and youths during program design, delivery, and evaluation was only 28%. However, studies have demonstrated that such involvement is essential to the successful functioning and delivery of health services [19, 24]. Moreover, it has been reported that adolescents who actively participate in program design and implementation are more likely to use contraception, engage in safe sexual practices, and have more favorable attitudes toward sexuality [24].

Regarding the accessibility and effectiveness of services, over eighty percent (82%) and 90% of adolescents and youths respectively, believed the service to be accessible and effective. The characteristics that resulted in high accessibility were service affordability/ the provision of services free of charge (100%) and convenient hours of operation (99%). The level of accessibility reported in the current study was inconsistent with a Mexican study where adolescents believe that friendly services are mainly out of reach because of administrative and financial obstacles in the rules and regulations [19]. The possible difference in accessibility of services might be due to differences in the study area. The majority of health facilities assessed in the current study were from the nation's capital city where basic health services coverage is relatively higher.

It has been reported that having trained providers and youth-friendly strategies in health facilities is necessary to ensure the effectiveness of AYFH services. Our results demonstrated that the AYFH service was seen to be effective (90%). Specifically, almost all health providers (98%) got the required training to work with youths and provide services, and also nearly all (99%) health facilities have youth-friendly strategies. This finding is higher than the AYFHS effectiveness documented in Nigeria [21], Mexico [19], and South Africa [22]. The possible reason for the difference can be explained by the difference in the characteristics assessed to measure the effectiveness and also the location of health facilities. The higher effectiveness of the service reported in the current study can contribute to the delivery of the right health service and can positively impact the health of adolescents in the study areas.

The overall satisfaction with AYFH service in the study area was 25.3%. This finding is lower than the level of satisfaction reported in northeast Ethiopia (47.2%) [11], Arba Minch, southern Ethiopia (49.1%) [25], and South Africa (81.7%) [10]. The possible reason for the difference can be the difference in the satisfaction measurement used in these studies. For example, the current study used eleven items for measuring adolescents' and youths' satisfaction. Out of these, the respondents who scored ≥ 6 from 11 points on client satisfaction tools and three compulsory items namely waiting time to get the service, privacy, and feeling comfortable with the services provided were used to assess satisfaction based on experts' recommendations. These criteria appear to be one of the most significant characteristics that have a strong link with client satisfaction [20, 26]. Another possible explanation can be the differences in study settings and the socio-demographic and socio-economic characteristics of the study population. The majority of participants in this study (81.2%) were female, and their median age (IQR) was more than 21 (5), which is significantly higher than the average age and sex of adolescents and youths found in prior studies conducted in southern and northeast Ethiopia [2, 25].

Age was found to be a significant predictor of AYH service satisfaction in our study; for every unit increase in the age of adolescents and youths, the likelihood of being satisfied with AYFH services dropped by 30%. This finding is consistent with a study done in southern Ethiopia [25] and Germany [8]. According to a study done in Germany, younger teenagers reported higher levels of satisfaction than older ones [8]. This might be explained by the fact that older adolescents have greater health concerns, have more sophisticated understanding and cognitive abilities, and start to formulate autonomous opinions about their social environments may contribute to a decrease in age satisfaction. However, a Mongolian study reveals that younger adolescents are less satisfied. This possible reason can be differences in service expectations, as well as the type and delivery of health services for adolescents [25, 27].

The study documented that female adolescents and youths were 89% less likely to be satisfied with AYFH services compared to males (AOR: 0.11, 95% CI: 0.04–0.33). This finding conflicts with research conducted in Dessie town, Ethiopia, which found that female adolescents had higher satisfaction with AYFH services [28]. Male has a stronger correlation with young-friendly SRH service consumption, particularly in Ethiopia, which may account for their greater levels of satisfaction with the service [1, 29]. This is because men are free to engage in society and are less likely to face judgment from medical professionals [1]. A low incidence of AYFH service utilization may be linked to varying expectations of healthcare services and a lack of knowledge about certain healthcare services, both of which could lower satisfaction levels.

Additionally, the study showed that adolescents and youth who had no prior information about AYFH services had 80% lower odds of satisfaction with AYFH services (AOR: 0.20, 95% CI: 0.10–0.44). This finding is in line with a systematic review of the quality of AYFH services, where those adolescents and youths who were informed about AYFH services had 1.6 times

higher satisfaction [30]. This is consistent with the fact that access to information is essential for making use of the services and ensuring client satisfaction [18].

Furthermore, the study revealed that a higher number of sexual partners was associated with decreased odds of satisfaction with AYFH services (AOR: 0.37, 95% CI: 0.21–0.65). The possible reason can be adolescents who have a higher number of sexual partners might be reluctant to open up to healthcare providers about their experiences, which makes it more difficult for them to build trust and establish harmonious relationships. Moreover, differences in the adolescent's behaviour, healthcare providers' attitude towards adolescents with multiple sexual partners [1], type of services requested, and services provided to adolescents [30] could also result in a disparity in satisfaction. For instance, a US study found that teenagers with several sexual partners have a greater likelihood of substance use [31].

### Strength of the study

Two main strengths of our study are: (i) Using data from a representative sample of adolescent users, we showed the degree to which AYFHS adheres to the standard of quality of care under the WHO friendliness domains. The WHO guidelines are often based on extensive research and evidence-based practices. The utilization of the guideline to measure youth-friendliness in health services promotes consistency, evidence-based practices, and quality improvement, ultimately leading to better health outcomes for young people (ii) We have measured client satisfaction using factors that significantly influence how young people use services, including waiting time to get the service, privacy, and feeling comfortable with the services provided as a mandatory criterion for satisfaction. To better align services with youth preferences and to increase trust and engagement with healthcare services, client satisfaction must be measured using factors that have a significant impact on how young people use services.

### Limitations of the study

This study was restricted to looking at the quality of services/care friendliness from the users' perspective. However, interviews with AYFHS health managers or healthcare professionals were not conducted for this study, which would have provided important information on supply-side perceptions of AYFHS quality. The use of exit interviews as sources of client satisfaction has another limitation in that they are conducted with a sample of clients who have already decided to visit a particular healthcare facility and are likely under the assumption that it will be at least satisfactory. Moreover, because the data analyzed were self-reported by adolescents and youths, social desirability and recall bias are possible.

### Conclusion

This study has highlighted the AYFH service exposure of adolescents, care friendliness, and their satisfaction with the services provided in Addis Ababa city administration and East Shewa zone in the Oromia region, Ethiopia. The quality of AYFH services provided at the health facilities did not meet all the five components of AYFHS, as nearly two-thirds of adolescents and youths perceived the AYFHS in the study settings to be inequitable and inappropriate. Additionally, adolescents and young people thought that AYFH services were moderately accessible and had a low acceptability rate. On the other hand, among the five AYFHS components, high effectiveness is documented. Overall client satisfaction with AYFH service in Addis Ababa city and the East Shewa zone of Oromia, Ethiopia was low. Variables such as age, sex, number of sexual partners, and prior information about AYFH services were significantly associated with client satisfaction with adolescent and youth-friendly health services.

To improve the standard of AYFHS offered at the healthcare facilities, relevant policy-makers and health managers should develop and properly implement policies that do not impose any limitations on healthcare provision and provide the required healthcare package to fulfill the needs of all adolescents. It is also essential to allow adolescents to share their experiences obtaining health services and express their needs and preferences. Furthermore, future research should look into evaluating the quality of services using an observation checklist for healthcare facilities and directly assessing the healthcare from the perspective of providers and managers. Lastly, to enhance adolescents' and youth's satisfaction with AYFH services, female and older adolescents and youths, information about AYFH services and a higher number of sexual partners should be targeted for intervention.

## Supporting information

**S1 File. English version of the questionnaire.**
(PDF)

**S1 Table. Measurement/scoring sheet/ for the five domains of care friendliness.**
(PDF)

## Acknowledgments

We are grateful to all adolescents and youths who participated in this study. We would like to express our gratitude to health managers, health workers, and data collectors in all participating health facilities. We thank Dorothy L. Southern for providing scientific writing advice and critically reviewing the manuscript.

## Author Contributions

**Conceptualization:** Andamlak Gizaw Alamdo, Fanna Adugna Debelle, Tewodros Getinet Yirtaw.

**Data curation:** Andamlak Gizaw Alamdo.

**Formal analysis:** Andamlak Gizaw Alamdo, Tewodros Getinet Yirtaw.

**Investigation:** Andamlak Gizaw Alamdo.

**Methodology:** Andamlak Gizaw Alamdo, Tewodros Getinet Yirtaw.

**Supervision:** Adom Manu, Juliana Yartey Enos.

**Writing – original draft:** Andamlak Gizaw Alamdo, Fanna Adugna Debelle, Peter Muriuki Gatheru.

**Writing – review & editing:** Andamlak Gizaw Alamdo, Fanna Adugna Debelle, Peter Muriuki Gatheru, Adom Manu, Juliana Yartey Enos, Tewodros Getinet Yirtaw.

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
