## [Decision Letter · Decision Letter 0]

13 May 2024

PONE-D-24-06588Youth-friendly health service in Ethiopia: Assessment of care friendliness and user’s satisfactionPLOS ONE

Dear Dr. Alamdo,

Thank you for submitting your manuscript to PLOS ONE. After careful consideration, we feel that it has merit but does not fully meet PLOS ONE’s publication criteria as it currently stands. Therefore, we invite you to submit a revised version of the manuscript that addresses the points raised during the review process.

**ACADEMIC EDITOR: ** Please provide adequate justification how the study areas (Addis Ababa city administration and East shewa zone of Oromia region) are selected. To make more informative and ensure replicability, reorganize methods section in the following way:Study design, setting and periodStudy participants and selection procedure (the study participants; sample size; sampling techniques; inclusion/exclusion criteria employed) Data collection tools and procedures (describe contents of the tool, whether tools are adapted or developed, sources, methods of data collection, data collectors and supervisors [background and qualifications, any training given]; upload the English version of the questionnaire as supplementary material)Study variables (definition and measurement of the variables)Data quality control Data processing and analysis Ethical consideration Figures should be self-explanatory (add axis titles); describe all abbreviations used within tables as table footnotes Discuss both strength and limitations of the study

We look forward to receiving your revised manuscript.

Kind regards,

Dawit Wolde Daka

Academic Editor

PLOS ONE

Journal Requirements:

Reviewers' comments:

Reviewer's Responses to Questions

**Comments to the Author**

1. Is the manuscript technically sound, and do the data support the conclusions?

Reviewer #1: Yes

Reviewer #2: Yes

2. Has the statistical analysis been performed appropriately and rigorously? 

Reviewer #1: Yes

Reviewer #2: No

3. Have the authors made all data underlying the findings in their manuscript fully available?

Reviewer #1: No

Reviewer #2: Yes

4. Is the manuscript presented in an intelligible fashion and written in standard English?

Reviewer #1: Yes

Reviewer #2: Yes

5. Review Comments to the Author

Reviewer #1: Reviewer comment to author

1. The study presents the results of primary scientific research. -yes

2. Results reported have not been published elsewhere.-yes

3. Experiments, statistics, and other analyses are performed to a high technical standard and are described in sufficient detail.-yes

4. Conclusions are presented in an appropriate fashion and are supported by the data-yes.

5. The article is presented in an intelligible fashion and is written in standard English-moderate/requires further modification.

6. The research meets all applicable standards for the ethics of experimentation and research integrity-yes/requires some details about waiver of assent for age under 18.

7. The article adheres to appropriate reporting guidelines and community standards for data availability-yes/but authors must submit STROBE checklist for cross sectional study.

General comments

Unnecessary use of words like “we” in page 2 line 31-32. Revise it.

What was the research gap regarding intention of women to give birth at health facility? If any put your strong justification in final paragraph of the background.

In background part you should describe YFS and YFS satisfaction briefly.

The problem was not well stated in introduction part. What was problem of friendliness? What was the problem of satisfaction? Consequences of poor satisfaction?

Describe the characteristics of study population regarding youth friendly services and health facility availability/accessibility, trained health care provider availability and other important information in your study setting.

I’m not satisfied with selection criteria for selecting one health center from east showa zone? Can we say it representative for all other health facilities in the zone?

Study period must be specified. Specific date of commencement and final date of data collection was not specified. You said from August to October 2022. See page 6 line 100

Measurement of accessibility, equity, acceptability, appropriateness,effectiviness must be clearly mentioned.

There was a problem of organizing your manuscript with appropriate sequence. For example sample size determination and sampling procedure must be organized before data collection procedure. Revise it.

You haven’t used supervisors? Why?

What method was used to control data quality?

Have you checked multi-collinearity of independent variables?

Have you checked model fittnesss?

Reccomendation must be specific and must address significant variables. But your recommendation was too general. See page 27 line 481-486.

Overall comment

Manuscript was well written and the problem under the study was interesting public health issue. My concern was justification for this study because the problem must be well stated. Measurement of accessibility, equity, acceptability, appropriateness,effectiviness must be clearly mentioned. Some editorial, topographic and description problem should be corrected.

Final decision- Accept with minor modification.

Reviewer #2: Title: “Youth-friendly health service in Ethiopia: Assessment of care friendliness and user’s satisfaction”

Thank you for inviting me the paper to review. The manuscript assessed friendliness and satisfaction level and their factors through a cross sectional study. Although quality and satisfaction of AYHS could be addressed under many qualitative and quantitative studies of “utilization of AYHS, the problem is still more contextual and may provide additional information to scientific and policy makers. But, the paper may benefit from revising the following comments.

Background

• Your justification “Even though 89 AFHS has been providing care in Ethiopia for nearly 20 years, there is a shortage of scientific research on the level of care that meets international standards” need to be more argued. Because there are many quantitative and qualitative why adolescents were not utilizing the AYFS, indirectly the issue of quality and satisfaction. What would this study makes a difference over those studies?

Methods

• How 50% of adolescents are assumed to be satisfied where there are many studies exploring many challenges why adolescents were not using the service. Could it be fair to assume there is no study while there are many studies? And, dependent variables are two in this study, “friendliness” and “satisfaction level”. Did you have any other justification why you did calculate separate sample size and compare the two to select the larger sample size?

• You selected health facilities first, then systematically selected adolescents and the study is also facility based, therefore, there could be larger variance. Why you were not considering other sample size determination techniques like design effect to increase your sample?

• Could the study be generalizable? Because usually most awarded adolescents are usually come frequently and they could have adequate knowledge about the characteristics of your interest. But, majority may not! Don’t you think that this is your limitation of the study?

• In your analysis, is it fair all the items in each construct could have reasonable factor loading despite easy to assume that they have reasonable reliability coefficients?

• “Many adolescents and youths aged less than 18 years utilize AYFH services without consulting their parents/guardians and asking for parental consent might cause emotional harm to adolescents and youths. Due to this, parents’ consent was not sought, and it was waived by the IRB of SPHMMC.” Could the IRB of SPHMMC waive anyone’s harm or benefit or autonomous? Parents should have been convinced or otherwise it could have been better excluded those under eight adolescents than justifying this way?

Results section

• The subheading “Demographic and socio-economic background of study participants” could have been shortened.

• In table 5, how one could know how many variables were taken to the multivariable logistic regression, and which of those are not associated ?

6. PLOS authors have the option to publish the peer review history of their article (what does this mean?). If published, this will include your full peer review and any attached files.

Reviewer #1: **Yes: **Elias Amaje Hadona

Reviewer #2: **Yes: **Yitagesu Habtu

---

## [Author Response · Author response to Decision Letter 0]

1 Jun 2024

Rebuttal Letter

Academic Editor

PLOS ONE

June 1, 2024

Dear academic editor,

Thank you for inviting us to submit a revised draft of our manuscript entitled, "Youth-friendly health service in Ethiopia: Assessment of care friendliness and user’s satisfaction" to PLOS ONE. We also value the time and energy you and the other reviewers invested in providing thoughtful feedback on how to strengthen our article. We are therefore delighted to resubmit our article for further consideration. We have made the required modifications in light of your extensive suggestions. We hope our edits and responses below satisfactorily address all the issues and concerns you and the reviewers have noted.

To facilitate your review of our revisions, the following is a point-by-point response to the questions and comments in a letter dated June 1, 2024

ACADEMIC EDITOR’S COMMENTS

First, thank you for your precious time, insightful comments, and suggestions for improving our paper. 

1. Academic editor’s comment: Provide adequate justification for how the study areas are selected. 

• Response: We value the time you took to fairly evaluate our methods section and point out any gaps. The study setting has been modified and the reason for selecting Addis Ababa city administration and East Shewa zone has been included in the document. The main reasons for selecting the study setting were, of course, the presence of functional youth-friendly health service corners in the health facilities and accessibility of the health facilities.

2. Academic editor’s comment: To make it more informative and ensure replicability, reorganize methods. 

• Response: Thank you for suggesting a way to reorganize the methods section. We reorganized the methods as follows: 

Study design, setting, and period

Study participants and selection procedure (inclusion/exclusion criteria are incorporated)

Sample size and sampling techniques 

Data collection tools and procedures (the contents of the tool are described, the methods of data collection, data collectors and supervisors, and their qualifications are also included, the English version of the questionnaire is uploaded as supplementary material)

Study variables (the measurement of the five domains of care-friendliness including equity, accessibility, acceptability, appropriateness, and effectiveness prepared in a separate document and uploaded as supplementary material (to manage the space in the methods part)). 

Data quality control 

Data processing and analysis 

Ethical consideration 

3. Academic editor’s comment: Figures should be self-explanatory (add axis titles); describe all abbreviations used within tables as table footnotes 

• Response: We have revised the title of our figures by adding the axis titles. Additionally, a description and a footnote are included for each abbreviation used within the tables 

4. Academic editor’s comment: Discuss both the strengths and limitations of the study

• Response: Thank you for this suggestion. We have made revisions in both parts. 

5. Academic editor’s comment: Include the following items (a rebuttal letter, a marked-up copy of your manuscript, and an unmarked version of your revised paper) when submitting your revised manuscript

• Response: Thank you for your suggestion. We have prepared all three documents as per the request and uploaded them as separate files

REVIEWER 1 COMMENTS: 

We sincerely appreciate you taking the time to read our paper, giving us insightful feedback, and providing all useful comments. 

6. Reviewer 1 comment: Unnecessary use of words like “we” in page 2 line 31-32. Revise it.

• Response: Thank you for your fair assessment. We have revised this part accordingly. 

7. Reviewer 1 comment: What was the research gap regarding the intention of women to give birth at a health facility? If any put your strong justification in the final paragraph of the background.?????

• Response: Thank you for raising this question. We have modified the background section indicating the gap in evidence about our study. However, we believe that the intention of women to give birth at the facility does not directly align with our main research question. 

8. Reviewer 1 comment: In the background part, you should describe YFS and YFS satisfaction briefly.

• Response: Thank you for this suggestion. We have explained AYFHS and Satisfaction with AYFH services in the background section. 

9. Reviewer 1 comment: The problem was not well stated in the introduction part. What was the problem of friendliness? What was the problem of satisfaction? Consequences of poor satisfaction?

• Response: Thank you for making a fair assessment of this section. In addition to explaining the AYFH services, satisfaction, and friendliness of services, we have indicated the existing gaps in friendliness, satisfaction, and the consequences of low satisfaction among adolescents and youths with AYFH services. In short, we have stated the problem in a separate paragraph under the background section. 

10. Reviewer 1 comment: Describe the characteristics of the study population regarding youth-friendly services and health facility availability/accessibility, trained health care provider availability, and other important information in your study setting.

• Response: Thank you for this suggestion. We have further modified the study setting, and now the study area's administrative layer, the overall population, the total adolescent population, and the accessibility of functional health facilities that provide services to youths and adolescents are all explained.

11. Reviewer 1 comment: I’m not satisfied with the selection criteria for selecting one health center from the east Shewa zone. Can we say it is representative of all other health facilities in the zone?

• Response: This is an interesting query. Modifications in the methods part particularly the study setting is made. We also believe that a single study site in East Shew cannot represent all the health facilities in the area. The rationale/reason for selecting the east Shewa zone of the Oromia region is explained.

12. Reviewer 1 comment: The study period must be specified. The specific date of commencement and final date of data collection was not specified. You said from August to October 2022. See page 6 line 100

• Response: Thank you for making a fair assessment of this part. The study was conducted from August 1 to October 28, 2022, and this is presented in the study period part of the paper.

13. Reviewer 1 comment: Measurement of accessibility, equity, acceptability, appropriateness, and effectiveness must be clearly mentioned

• Response: Thank you for this suggestion. The measurement of the five domains of care-friendliness including equity, accessibility, acceptability, appropriateness, and effectiveness prepared in a separate document and uploaded as supplementary material (to manage the space in the methods part). 

14. Reviewer 1 comment: There was a problem of organizing your manuscript with the appropriate sequence. For example, sample size determination and sampling procedure must be organized before the data collection procedure. Revise it.

• Response: Thank you for your suggestion. Following your and the academic editor's suggestions, we reorganized the methods section (please see the complete response provided under the academic editor's comment 2 above.)

15. Reviewer 1 comment: You haven’t used supervisors? Why? What method was used to control data quality?

• Response: Thank you for raising this question. We have supervisors as members of the research team for the study. We have added a data quality control section under the methods part. The number, the qualifications, and the role of data collectors and supervisors are explained. 

16. Reviewer 1 comment: Have you checked the multi-collinearity of independent variables? Have you checked model fitness?

• Response: You have raised an important question. We did not check this statistically/ using tests. However, we had to observe the variables to check whether there was such suspicion. None of the variables are suspected of that. In addition, the regression analysis supports our claim, by not resulting in estimates with wider CIs and or larger ORs. Regarding the model fitness, we have run the final model in Stata for HL and checked the model fitness. 

17. Reviewer 1 comment: The recommendation must be specific and must address significant variables. But your recommendation was too general. See page 27 line 481-486.

• Response: Thank you for this suggestion. We have changed the recommendation part as per the suggestion. We also believe that equity and appropriateness that were mentioned in the recommendation part are general and more specific characteristics under equity and appropriateness are identified and incorporated in the document 

REVIEWER 2 COMMENTS: 

18. Reviewer 2 general comments: The manuscript assessed friendliness and satisfaction level and their factors through a cross sectional study. Although quality and satisfaction of AYHS could be addressed under many qualitative and quantitative studies of “utilization of AYHS, the problem is still more contextual and may provide additional information to scientific and policy makers. But, the paper may benefit from revising the following comments.

• Response: Thank you very much for reading our work, contributing informative comments, and all of your other valuable feedback.

19. Reviewer 2 comments [Background]: Your justification “Even though 89 AFHS has been providing care in Ethiopia for nearly 20 years, there is a shortage of scientific research on the level of care that meets international standards” need to be more argued. Because there are many quantitative and qualitative why adolescents were not utilizing the AYFS, indirectly the issue of quality and satisfaction. What would this study makes a difference over those studies?

• Response: Thank you for making a fair assessment of this part. The existing literature focuses mainly on examining factors that affect AYFH services utilization and examining quality of care concentrating only on a particular aspect. Therefore, this study provides local, context-specific evidence about the degree of friendliness provided by adolescent and youth health services in Ethiopia by using the WHO Quality Assessment Guidebook

20. Reviewer 2 comments [Methods]: How 50% of adolescents are assumed to be satisfied where there are many studies exploring many challenges why adolescents were not using the service. Could it be fair to assume there is no study while there are many studies? And, dependent variables are two in this study, “friendliness” and “satisfaction level”. Did you have any other justification why you did calculate separate sample size and compare the two to select the larger sample size?

• Response: Thank you for raising this question: We believe that this could be a limitation of the study. However, during the proposal and data collection stage, we couldn't find similar studies done in similar settings. In addition, assuming 50% would give us a larger sample size at a 5% margin of error relative to other prevalence. So, the current sample size covers any prevalence at the 5% margin of error. Having separate sample sizes is not okay while we assume 50%, it is corrected in the manuscript.

21. Reviewer 2 comments [Methods]: You selected health facilities first, then systematically selected adolescents and the study is also facility-based, therefore, there could be a larger variance. Why you were not considering other sample size determination techniques like design effect to increase your sample?

• Response: Thank you for raising this question: 

1. We selected the health facilities using the lottery method, assuming that they are homogeneous.

2. The sampling design is not that much complex.

3. Time feasibility, based on client flow it would take longer than required/feasible time to get larger samples/ than we considered.

4. Based on the rule of thumbs the sample size is more than enough to make an inference relative to the number of variables considered in the regression.

 Considering all of the above we did not consider the design effect.

22. Reviewer 2 comments [Methods]: Could the study be generalizable? Because usually most awarded adolescents are usually come frequently and they could have adequate knowledge about the characteristics of your interest. But, majority may not! Don’t you think that this is your limitation of the study?

• Response: You have raised an important question. We agree with you, it can be one of the limitations of the study. It is already presented in the document as follows: The use of exit interviews as sources of client satisfaction has limitations in that they are conducted with a sample of clients who have already decided to visit a particular healthcare facility and are likely under the assumption that it will be at least satisfactory. 

23. Reviewer 2 comments [Methods]: In your analysis, is it fair all the items in each construct could have reasonable factor loading despite easy to assume that they have reasonable reliability coefficients?

• Response: Thank you for raising this question. We have adopted the tool from the WHO and assumed that it is a validated tool and standard guide, which is why we did not check the factor loadings using CFA. 

24. Reviewer 2 comments [Methods]: Many adolescents and youths aged less than 18 years utilize AYFH services without consulting their parents/guardians and asking for parental consent might cause emotional harm to adolescents and youths. Due to this, parents’ consent was not sought, and it was waived by the IRB of SPHMMC.” Could the IRB of SPHMMC waive anyone’s harm or benefit or autonomous? Parents should have been convinced or otherwise it could have been better excluded those under eight adolescents than justifying this way?

• Response: Thank you for raising this question: Ethics is always in support of benefits and waiving is also the mandate of the ethics committee for the benefit of the study participants. In this scenario, waiving the consent for the minors outweighs the risk of excluding youth/letting them know their parents. As the problem still exists and excluding minors will cost a huge information gap, waiving was considered as the best option, considering the risks and benefits of waiving the consent form. 

25. Reviewer 2 comments [Results]: The subheading “Demographic and socio-economic background of study participants” could have been shortened.

• Response: Thank you for your suggestion. This is shorted as socio-demographic characteristics. 

26. Reviewer 2 comments [Results]: In Table 5, how one could know how many variables were taken to the multivariable logistic regression, and which of those are not associated?

• Response: Thank you for this suggestion. This part is modified, the number of variables that had a statistical association in the bivariate analysis are presented in the document and the number of variables included in the multivariate logistic regression is also described. 

Again, thank you for allowing us to strengthen our manuscript with your valuable comments and queries. We have worked hard to incorporate your feedback and hope that these revisions persuade you to accept our submission.

Sincerely,

Andamlak Gizaw Alamdo

Corresponding Author

Department of Health Service Management, Health Promotion, Reproductive Health, and Nutrition, School of Public Health, Saint Paul's Hospital Millennium Medical College, Addis Ababa, Ethiopia

gizandal@gmail.com/andamlak.gizaw@sphmmc.edu.et

+251912038993

---

## [Decision Letter · Decision Letter 1]

24 Jun 2024

PONE-D-24-06588R1Youth-friendly health service in Ethiopia: Assessment of care friendliness and user’s satisfactionPLOS ONE

Dear Dr. Alamdo,

Thank you for submitting your manuscript to PLOS ONE. After careful consideration, we feel that it has merit but does not fully meet PLOS ONE’s publication criteria as it currently stands. Therefore, we invite you to submit a revised version of the manuscript that addresses the points raised during the review process.

ACADEMIC EDITOR: Up on my own review of the revised manuscript, there are some issues that needs clarifications. Pay attention and adequately address the points indicated in the manuscript file.

Kind regards,

Dawit Wolde Daka

Academic Editor

PLOS ONE

Journal Requirements:

Reviewers' comments:

Reviewer's Responses to Questions

**Comments to the Author**

1. If the authors have adequately addressed your comments raised in a previous round of review and you feel that this manuscript is now acceptable for publication, you may indicate that here to bypass the “Comments to the Author” section, enter your conflict of interest statement in the “Confidential to Editor” section, and submit your "Accept" recommendation.

Reviewer #1: All comments have been addressed

Reviewer #2: All comments have been addressed

2. Is the manuscript technically sound, and do the data support the conclusions?

Reviewer #1: Yes

Reviewer #2: Yes

3. Has the statistical analysis been performed appropriately and rigorously? 

Reviewer #1: Yes

Reviewer #2: Yes

4. Have the authors made all data underlying the findings in their manuscript fully available?

Reviewer #1: Yes

Reviewer #2: Yes

5. Is the manuscript presented in an intelligible fashion and written in standard English?

Reviewer #1: Yes

Reviewer #2: Yes

6. Review Comments to the Author

Reviewer #1: All comments were addressed adequately. this manuscript is suitable for publication in plos one journal fulfilling all criteria.

Reviewer #2: (No Response)

7. PLOS authors have the option to publish the peer review history of their article (what does this mean?). If published, this will include your full peer review and any attached files.

Reviewer #1: **Yes: **Elias Amaje Hadona

Reviewer #2: **Yes: **Yitagesu Habtu

---

## [Author Response · Author response to Decision Letter 1]

27 Jun 2024

Academic Editor

PLOS ONE

Date: June 27, 2024

Author’s response to reviews: Version _2

Title: Youth-friendly health service in Ethiopia: Assessment of care friendliness and user’s satisfaction

Academic editor’s comments 

Many Thanks. We have learnt from your comments, and have tried our best to address them. 

Academic editor’s comment [Abstract]: Is AOR be used to declare the presence of statistically significant associations? Rewrite this description. 

• Response: Thank you for pointing this out. We have corrected it. 

Academic editor’s comment [Methods]: Use the most updated/recent population size of the study areas. Currently, Addis Ababa’s population is more than 5 million.

• Response: We noticed the oversight and have corrected it. We have come in more recent data, and have updated the population size accordingly.

1. Academic editor’s comment [Methods]: The assumption of including this zone, over other zones/districts all over the country is no sufficiently justified. Strongly recommend to briefly put the explanations of including this district, and no other districts with the service (AYFHS).

• Response: We appreciate your point. The criteria considered when choosing the East Shewa zone for the study were briefly described again. 

2. Academic editor’s comment [Methods]: Cite the literature reviewed. 

• Response: Thank you. We have cited the references reviewed. 

3. Academic editor’s comment [Methods]: Is this the questionnaire? Use correct labels while supplementary materials. 

• Response: Yes, it’s the questionnaire. We have revised the label for supplementary material 

4. Academic editor’s comment [Methods]: Merge this with the operational definitions section. You can name as: Study variables 

• Response: Thank you for your suggestion. We have merged the study variables and operational definition as ‘Study variables”

5. Academic editor’s comment [Methods]: OR is measure of strength of association, it is only p-value and 95% CI that is used to declare the presence of statistically significant associations. Correct descriptions accordingly

• Response: Thank you, we noticed the error. We revised it accordingly. 

6. Academic editor’s comment [Methods]: Indicate the date as well under the ethical consideration sub-section. 

• Response: Thank you. We have incorporated the date and revised a paragraph. 

7. Academic editor’s comment [Results, Table 1]: The way age variable presented is not clearer, suggest using categories. It is a district, not Oromia. Correct accordingly. Are both outputs similar? Use mean with SD. Suggest categorizing age, and describe the mean results in narrations.

• Response: Thank you for this suggestion. We have revised this part accordingly. We categorized the age, presented it in the table, and omitted the mean and median results already narrated in the report. 

8. Academic editor’s comment [Results, Table 2-3]: Describe all acronyms/abbreviations used as table footnote. Try to concise variable names. See highlighted ones and correct. You can describe the details in narration. 

• Response: Revised based on the suggestions. Variable names are revised and the mean and median results are deleted from the table.

9. Academic editor’s comment [Results, Table 5]: It would have been good if the reference group be reversed (Factors associated with adolescent and youth client satisfaction). 

• Response: Thank you for raising this concern and we agree with you. It could have been better if the reference category had a larger frequency. However, during data analysis, we have considered other issues such as consistency with previous research, and interpretability (to have a meaningful comparison). 

10. Academic editor’s comment [Results, Table 5]: What does this (‘1’) represent? Describe as table footnote. Describe all other abbreviations used in the table.

• Response: Thank you. "1’: indicates the reference and the baseline category for the binary and multivariable logistic regression analysis, respectively. All abbreviations/acronyms are described as footnotes. 

1. Reviewer 1 Comments: All comments were addressed adequately. this manuscript is suitable for publication in PLoS One journal fulfilling all criteria.

• Response: Thank you. Your input has been very useful in enhancing the paper.

2. Reviewer 1 and Reviewer 2: Overall comments

Response: Many thanks for your support in enhancing the quality of our paper. 

Final comment: 

Journal Requirements: Please review your reference list to ensure that it is complete and correct. 

Response: Thank you. We have reviewed our reference list and revised it thoroughly according to the journal requirements.

---

## [Editor Report · Decision Letter 2]

2 Jul 2024

Youth-friendly health service in Ethiopia: Assessment of care friendliness and user’s satisfaction

PONE-D-24-06588R2

Dear Dr. Alamdo,

We’re pleased to inform you that your manuscript has been judged scientifically suitable for publication and will be formally accepted for publication once it meets all outstanding technical requirements.

Kind regards,

Dawit Wolde Daka

Academic Editor

PLOS ONE
---

## [Editor Report · Acceptance letter]

8 Jul 2024

PONE-D-24-06588R2 

PLOS ONE

Dear Dr. Alamdo, 

I'm pleased to inform you that your manuscript has been deemed suitable for publication in PLOS ONE. Congratulations! Your manuscript is now being handed over to our production team.

Kind regards, 

on behalf of

Mr Dawit Wolde Daka 

Academic Editor

PLOS ONE